# Chemical Properties, Preparation, and Pharmaceutical Effects of Cyclic Peptides from *Pseudostellaria heterophylla*

**DOI:** 10.3390/molecules30122521

**Published:** 2025-06-09

**Authors:** Yue Yang, Luan Wen, Zhuang-Zhuang Jiang, Ben Chung-Lap Chan, Ping-Chung Leung, Chun-Kwok Wong, Ning-Hua Tan

**Affiliations:** 1State Key Laboratory of Natural Medicines, Department of TCMs Pharmaceuticals, School of Traditional Chinese Pharmacy, China Pharmaceutical University, Nanjing 211198, China; 3323020993@stu.cpu.edu.cn (Y.Y.); 3121020127@stu.cpu.edu.cn (Z.-Z.J.); nhtan@cpu.edu.cn (N.-H.T.); 2Institute of Chinese Medicine, The Chinese University of Hong Kong, Hong Kong, China; 1155209219@link.cuhk.edu.hk (L.W.); pingcleung@cuhk.edu.hk (P.-C.L.); ck-wong@cuhk.edu.hk (C.-K.W.); 3Department of Chemical Pathology, Faculty of Medicine, The Chinese University of Hong Kong, Hong Kong, China; 4Li Dak Sum Yip Yio Chin R & D Centre for Chinese Medicine, The Chinese University of Hong Kong, Hong Kong, China

**Keywords:** *P. heterophylla*, cyclic peptides, biological activities

## Abstract

*Radix Pseudostellariae* (Tai-Zi-Shen), the dried tuberous root of the Caryophyllaceae plant *Pseudostellaria heterophylla* (Miq.) Pax (*P. heterophylla*), has been widely used in traditional Chinese medicine (TCM) for thousands of years. It is slightly bitter, neutral in nature, associated with the spleen and lung meridians, and used for nourishing qi, invigorating the spleen, as well as promoting body fluid production and moistening the lungs. In recent years, with the development in natural product chemistry, cyclic peptides, as some of the active constituents derived from *P. heterophylla,* have gained increasing attention. These cyclic peptides demonstrate a broad range of biological activities, including anticancer, antioxidant, and immunomodulatory effects, as well as cognitive benefits. This review provides an overview of the chemical characteristics and preparation strategies of cyclic peptides from *P. heterophylla*, and their biological activities and potential mechanisms are also described. The presented work establishes a scientific framework to facilitate the future research and development of *P. heterophylla* cyclic peptides as potential therapeutic agents.

## 1. Introduction

*Radix Pseudostellariae* (Tai-Zi-Shen), the dried tuberous root of the Caryophyllaceae plant *P. heterophylla*, is mainly distributed in Jiangsu, Fujian, Guizhou, and Anhui [1,2]. According to the theory of traditional Chinese medicine (TCM), *P. heterophylla* has a mildly bitter taste and exhibits neutral pharmacological properties, primarily targeting the spleen and lung systems. *P. heterophylla* assists in maintaining physiological functions and replenishing body fluids while improving the functional activities of the spleen and lungs. It is commonly employed in clinical practice to address symptoms such as anorexia, fatigue, and dry cough. Phytochemical investigations have revealed that *P. heterophylla* contains a variety of bioactive components, including cyclic peptides, polysaccharides, saponins, and amino acids, among which cyclic peptides with distinctive cyclic structures are its characteristic constituents [3].

Cyclic peptides are molecules with a closed-ring structure formed by amino acids linked through peptide bonds, and they are widely found in animals, plants, and microorganisms in nature. Compared to general linear peptides, cyclic peptides have more stable and uniform conformations with unique structures, which confer better resistance to chemical degradation and enzymatic hydrolysis [4]. Modern pharmacological studies have shown that most cyclic peptides exhibit activities characterized by prolonged half-lives, high efficiency, strong metabolic stability, high bioavailability, and strong receptor selectivity [5]. These compounds exhibit significant biological activities, such as insecticidal [6], antimicrobial [7], anticancer [8], uterotonic [9], and immune-enhancing effects [10].

Compared to other plant-derived cyclic peptides such as cyclotides and orbitides, cyclic peptides from *P. heterophylla* exhibit relatively simple structures that facilitate chemical or biosynthetic production [11], and demonstrate various pharmacological activities, including anti-tumor, anti-inflammatory, antioxidant, immune-modulatory, and memory-enhancing activities. This review systematically summarizes the chemical properties, preparation strategies, biological activities, and potential mechanisms of *P. heterophylla* cyclic peptides.

## 2. Chemical Features of Cyclic Peptides from *P. heterophylla*

Cyclic peptides are polypeptide chains with cyclic ring structures. The amide or other chemically stable bonds link the end of the peptide and the other, forming a ring structure with enormous structural diversity [12]. Cyclic peptides are considered an important source of drugs derived from natural products, and more than 40 cyclic peptides have been approved for clinical applications [13]. Cyclic peptides exhibit a wide range of biological activities, and the interaction between the unique cyclic backbone and cell membranes might be the primary mechanism [14]. In addition to membrane permeation, conformational freedom is reduced because of the cyclization of peptides, which enhances metabolic stability. Meanwhile, compared with typical small molecules (Mw < 500 Da), the larger molecular size of most commonly used cyclic peptides (6–15 aa; Mw = 500–2000 Da) might have advantages in binding proteins with better receptor specificity [15].

Cyclic peptides represent a prominent class of bioactive constituents in *P. heterophylla*. Current phytochemical investigations have led to the isolation and characterization of 19 distinct cyclic peptides from this medicinal plant (Table 1, Figure 1), which are categorized into two major structural classes: 9 heterophyllin-type peptides and 10 pseudostellarin-type peptides.

In 1993, Tan et al. [16] isolated heterophyllin A (HA) and heterophyllin B (HB) from *P. heterophylla*, marking the beginning of research on cyclic peptides from this species. Subsequently, two additional novel compounds—heterophyllin C (HC) [17] and heterophyllin J (HJ) [18]—were obtained. Morita et al. [19,20,21] sequentially isolated and identified eight structurally novel cyclic peptides from *P. heterophylla*, named pseudostellarin A–pseudostellarin H (PA–PH). In traditional medicine, the fibrous roots of *P. heterophylla* are typically discarded as waste. In 2022, Zhao et al. [3] isolated and identified pseudostellarin K (PK), a novel cyclic hexapeptide, from the fibrous roots of *P. heterophylla*. Moreover, Chen et al. [22] employed various chromatography methods and spectroscopic techniques to isolate and identify a novel cyclic pentapeptide, named pseudostellarin L (PL), from the ethyl acetate fraction of a 75% ethanol extract of the fibrous roots of *P. heterophylla*.

**Table 1 molecules-30-02521-t001:** Summary of cyclic peptides from *P. heterophylla*.

No.	Compound	Ring Size (aa)	Structure	Mw (*m/z*)	Ref.
1	Heterophyllin A	7	Cyclo(Thr-Pro-Val-Ile-Phe-Gly-Ile)	727.9	[16]
2	Heterophyllin B	8	Cyclo(Gly-Gly-Leu-Pro-Pro-Pro-Ile-Phe)	778.9	[16]
3	Heterophyllin C	7	Cyclo(Gly-Pro-Ile-Ile-Pro-Ile-Leu)	703.9	[17]
4	Heterophyllin D	6	Cyclo(Gly-Phe-Ile-Thr-Val-Phe)	664.8	[23]
5	Heterophyllin E	10	Cyclo(Val-Tyr-Ala-Gly-Pro-Tyr-Leu-Ala-Gly-Pro)	989.1	[23]
6	Heterophyllin F	6	Cyclo(Ile-Ile-Leu-Leu-Leu-Gly)	622.8	[23]
7	Heterophyllin G	7	Cyclo(Pro-Val-Ile-Phe-Gly-Ile-[Thr-O(CH_2_)_4_CH_3_])	798.0	[23]
8	Heterophyllin H	2	Cyclo(Tyr-Pro)	260.3	[23]
9	Heterophyllin J	5	Cyclo(Ala-Gly-Pro-Val-Tyr)	487.6	[18]
10	Pseudostellarin A	5	Cyclo(Gly-Pro-Tyr-Leu-Ala)	501.6	[19]
11	Pseudostellarin B	8	Cyclo(Gly-Ile-Gly-Gly-Gly-Pro-Pro-Phe)	682.8	[19]
12	Pseudostellarin C	8	Cyclo(Gly-Thr-Leu-Pro-Ser-Pro-Phe-Leu)	812.9	[19]
13	Pseudostellarin D	7	Cyclo(Gly-Gly-Tyr-Pro-Leu-Ile-Leu)	713.9	[20]
14	Pseudostellarin E	9	Cyclo(Gly-Pro-Pro-Leu-Gly-Pro-Val-Ile-Phe)	878.1	[20]
15	Pseudostellarin F	8	Cyclo(Gly-Gly-Tyr-Leu-Pro-Pro-Leu-Ala-Pro)	784.9	[20]
16	Pseudostellarin G	8	Cyclo(Phe-Ser-Phe-Gly-Pro-Leu-Ala-Pro)	816.9	[21]
17	Pseudostellarin H	8	Cyclo(Gly-Thr-Pro-Thr-Pro-Leu-Phe-Phe)	861.0	[21]
18	Pseudostellarin K	6	Cyclo(Ile-Phe-Gly-Thr-Val-Phe)	664.8	[3]
19	Pseudostellarin L	5	Cyclo(Pro-Gly-Tyr-Phe-Val)	563.7	[22]

## 3. Preparation for Cyclic Peptides from *P. heterophylla*

Cyclic peptides are pervasive in microorganisms, plants, and animals. With the improved membrane permeation, stability, and receptor specificity, cyclic peptides exhibited more potent biological activity and superior bioavailability compared to linear peptides, and they have received increasing attention in recent years. They can be obtained through direct extraction, biosynthesis, and chemical synthesis to fulfill specific research or therapeutic needs [24]. Although the preparation of *P. heterophylla* cyclic peptides in early studies was mainly based on direct extraction and silica gel separation, more research has been conducted on synthetic methods afterwards to improve the preparation efficiency, including biosynthesis and chemical synthesis. In the view of biosynthesis, the cyclic peptides from *P. heterophylla* are considered ribosomally synthesized and post-translationally modified peptides (RiPPs), and both in vivo and in vitro cyclization strategies of *P. heterophylla* cyclic peptides were studied. As for chemical synthesis, the solid-phase peptide synthesis (SPPS) method is commonly used in recent studies.

### 3.1. Direct Extraction and Separation

To extract the cyclic peptides, the dried roots of *P. heterophylla* are usually extracted with polar organic solvents and then partitioned with other organic reagents of slightly lower polarity to obtain the corresponding fractions. For example, Tan et al. [16] extracted *P. heterophylla* by boiling 95% ethanol and further extracted the ethanol fraction with ethyl acetate. In 1994, Morita et al. [18,19,20] obtained the methanol extracts of *P. heterophylla* and partitioned them with n-butanol.

For further separation, the ethyl acetate or n-butanol fractions were purified by silica gel column chromatography [16,18], Sephadex LH-20, and ODS columns [25]. In addition to traditional separation methods, high-speed countercurrent chromatography (HSCCC) has been widely applied in the separation and purification of natural products in recent years due to its advantages, including the absence of irreversible adsorption, high recovery rate, and ease of operation [26]. For example, Han et al. [27] employed HSCCC to process *P. heterophylla* extracts, obtaining HB with a purity greater than 96%.

### 3.2. Biosynthesis

Although the direct isolation of cyclic peptides from *P. heterophylla* is a straightforward method, it is usually limited by poor yields and high production costs. The production of cyclic peptides from *P. heterophylla* using biosynthesis has thus become a better alternative in recent years.

In early biosynthesis studies, scientists mainly focused on the cyclization methods of linear peptides to produce *P. heterophylla* cyclic peptides. In 1999, Scott et al. [28] established an in vivo catalysis method for peptide backbone cyclization by using split inteins and successfully synthesized PF, a peptide with tyrosinase inhibitory activity, in *Escherichia coli*. They ligated DnaE split intein (IN and IC) from *Synechocystis* sp. PCC6803 to the target sequence of PF to construct a recombinant plasmid, which induced the cyclization of the linear peptides in *E. coli* to obtain PF, and the yield was 2–20 mg/L. In addition to intracellular methods, the in vitro cyclization methods for cyclic peptides from *P. heterophylla* were also studied. In 2006, Jia et al. [29] isolated a crude enzyme (PH-1) from *P. heterophylla* capable of catalyzing the enzymatic cyclization of linear peptide substrates into HB. After synthesizing the target linear peptide using the SPPS method, they performed an enzymatic reaction under optimized conditions to produce HB. Similarly, in 2012, Xu et al. [30] developed an in vitro culture system for HB biosynthesis. They chemically synthesized the linear peptide precursor of HB (LHB) and its N-acetyl cysteamine thioester (LHB-SNAC) and then prepared the cell-free enzyme extract (CFE) of *P. heterophylla*. Under specific conditions, a small amount of HB was produced upon incubating LHB-SNAC with the CFE, yielding 850 μg/g.

The biosynthesis of cyclic peptides mainly follows two distinct pathways. One involves non-ribosomal peptide synthetases, which assemble peptide chains in a template-independent manner. The other route involves the RiPP pathway, in which the linear precursor peptides are translated by the ribosome and then post-translationally modified to form cyclic structures [31]. To better understand the biosynthesis mechanism of cyclic peptides, it is essential to investigate the sequences and structural features of precursor peptides. In 2019, Zheng et al. [31] established a screening method for the precursor gene (*prePhHB*) of HB by using the *P. heterophylla* RNA-seq database. Both in vivo and in vitro experiments confirmed that *prePhHB* contributed to the precursor peptide production to synthesize HB, indicating that HB is enzymatically generated through the RiPP pathway. Furthermore, in 2024, Wu et al. [32] found that the methyl jasmonate treatment could increase the expression of *prePhHB* in *P. heterophylla* tuberous roots, thereby promoting HB biosynthesis. The transcription factors played a crucial role in this process.

### 3.3. Chemical Synthesis

In previous studies, two primary synthetic strategies were commonly employed for the chemical synthesis of cyclic peptides derived from *P. heterophylla*.

The first approach involves the direct coupling of short linear peptide fragments (e.g., dipeptides) to form extended linear peptides, followed by intramolecular cyclization to generate the target cyclic peptide. In 1999, Himaja et al. [33] developed a synthetic strategy for the cyclic heptapeptide PD through the sequential coupling of dipeptide and amino acid building blocks. Initially, two dipeptide units and one amino acid unit were prepared and progressively coupled to form linear dipeptides, tripeptides, tetrapeptides, and heptapeptides. The linear heptapeptide was then cyclized using the p-nitrophenyl ester method to obtain the cyclic heptapeptide PD, with a total yield of 76.4%. Similarly, different dipeptides were coupled to obtain tetrapeptides and octapeptides, which were then cyclized to produce PB [34] and PG [35], respectively, and the overall yields were 61.0% and 58.5%, respectively.

The second strategy employs SPPS to synthesize the linear precursor of the target cyclic peptides, which are then cyclized to yield the desired cyclic peptides. In 2016, Zhang et al. [36] employed Fmoc-SPPS to synthesize the target linear peptide precursor, which was subsequently cyclized into PA and HJ under specific conditions, with the total yields of 76% and 68%, respectively. In 2018, Liu et al. [37] explored a novel recyclable SPPS strategy based on the hypervalent iodine (III) reagent FPID and successfully applied this method to synthesize PD in 32% isolated yield (44% NMR yield).

## 4. Pharmacokinetic Characteristics

Pharmacokinetics is a bridge linking herbal medicines and their pharmacological responses [38]. Zhao et al. [39,40] developed and validated an LC-ESI-MS/MS method for the first time to achieve highly sensitive quantification of HB in rat plasma. In their study, SD rats were administered intravenous injections of HB at different doses (2.08, 4.16, and 8.32 mg/kg) via the tail vein. A cross-over sampling design was employed, with blood samples collected from 5 min to 7 h post-administration to assess the pharmacokinetic behavior of HB in vivo. The results demonstrated that HB exhibited linear pharmacokinetics and was rapidly eliminated from plasma. Huang et al. [41] established a UPLC-MS/MS method to investigate the pharmacokinetics of HB in rats. Following the intravenous administration of HB at 20 mg/kg in SD rats, plasma samples were collected at multiple time points to analyze the compound’s absorption, distribution, metabolism, and excretion profiles. The study found that HB was rapidly distributed and eliminated, with an extremely large volume of distribution, indicating widespread distribution into tissues and organs. The specific pharmacokinetic parameters of HB at different doses in SD rats are summarized in Table 2.

## 5. Biological Activities of Cyclic Peptides from *P. heterophylla*

Cyclic peptides derived from *P. heterophylla* exhibit significant pharmacological activities across a wide range of areas, including anticancer, immune modulation, antioxidation, anti-diabetic, anti-inflammatory effects, the regulation of gut microbiota, the enhancement of cognitive function, anti-fibrotic properties, and the inhibition of tyrosinase activity.

### 5.1. Anti-Tumor Activity

TCMs exhibit significant anti-tumor effects, not only alleviating the symptoms of tumor patients and controlling the tumor size but also enhancing the efficacy of chemotherapy by mitigating its side effects when used in combination with chemotherapeutic agents [42]. As the key bioactive component of *P. heterophylla*, cyclic peptides have demonstrated potent anticancer activity in both in vitro and in vivo studies (Table 3).

Shi et al. [43] investigated the regulatory effects of HB on ovarian cancer (OC) and explored its underlying mechanisms. After treatment with HB, the viability of OC cell lines (OVCAR8 and SKOV3) decreased in a concentration-dependent manner, with IC_50_ values of 67.88 and 78.27 μM, respectively. In vivo studies demonstrated that HB (20 mg/kg) significantly suppressed the growth of tumor xenografts in nude mice. These findings indicate that HB inhibited cell proliferation, migration, and invasion, and induced apoptosis in OC cells. Mechanistically, HB suppressed the malignant progression of OC cells by targeting and inhibiting the NRF2/HO-1 signaling pathway.

Xue et al. [44] explored the anti-gastric cancer (GC) effects of HB, with a particular focus on its role in activating endoplasmic reticulum (ER) stress in vitro and in vivo. Molecular characterization revealed a significant upregulation of key ER stress markers, including inositol-requiring enzyme 1 (IRE1), C/EBP homologous protein (CHOP), and glucose-regulated protein 78 (GRP78), accompanied by a downregulation of the anti-apoptotic protein B-cell lymphoma 2 (Bcl-2). ER stress inhibitor 4-phenylbutyric acid could partly reverse the suppressive effect of HB on GC cells. Furthermore, Wei et al. [45] investigated the pharmacological mechanisms of HB in GC cells. Overexpression of CXCR4 in GC cells significantly enhanced cell migration and invasion. HB was found to inhibit these malignant phenotypes by binding to CXCR4, thereby suppressing cell migration, invasion, and proliferation. The binding relationship between HB and CXCR4 was confirmed through cellular thermal shift assay and molecular docking. Furthermore, HB treatment reduced the phosphorylation levels of PI3K and AKT, indicating that HB inhibits the PI3K/AKT signaling pathway. HB also decreased PD-L1 expression, likely due to the suppression of the PI3K/AKT pathway. When CXCR4 was overexpressed, PD-L1 expression was restored. Thus, HB may exert its inhibitory effects by binding to CXCR4, suppressing the PI3K/AKT pathway, and reducing PD-L1 expression, thereby blocking tumor cell immune escape.

Tantai et al. [46] investigated the effects of HB on the adhesion and invasion of ECA-109 esophageal carcinoma (EC) cells, with a focus on the role of the PI3K/AKT/β-catenin signaling pathway. The study found that HB inhibited the expression of the PI3K/AKT pathway in ECA-109 cells, leading to a downregulation of β-catenin expression, which in turn suppressed cell migration and metastasis. The expression of downstream proteins, such as E-cadherin, was increased following HB treatment, while the expression of invasion-related factors, including snail, vimentin, MMP-2, and MMP-9, was significantly reduced. Therefore, HB may effectively suppress the adhesion and invasion of EC cells by downregulating the PI3K/AKT/β-catenin pathway and regulating its downstream molecules.

### 5.2. Anti-Inflammatory Activity

As shown in Table 4, studies have demonstrated that cyclic peptides of *P. heterophylla* possess anti-inflammatory activity in both in vitro and in vivo models.

Yang et al. [47] established an inflammatory injury model using lipopolysaccharide (LPS) in RAW 264.7 cells to investigate the anti-inflammatory and antioxidant activity of HB, along with its underlying mechanisms. The results revealed that HB significantly inhibited the production of nitric oxide (NO), interleukin-6 (IL-6), interleukin-1 beta (IL-1β), reactive oxygen species (ROS), and other inflammatory factors. This effect was mediated through modulating the PI3K/Akt signaling pathway, which also contributed to the inhibition of apoptosis.

AMPK serves as a common mediator, playing a crucial role in regulating cellular processes in different tissues by promoting autophagy, reducing oxidative stress, and alleviating pyroptosis. However, the specific downstream pathways and physiological responses vary across these distinct pathological conditions. For example, Zhang et al. [48] investigated the therapeutic efficacy of HB in spinal cord injury (SCI). Their results indicated that HB activated the AMPK-TRPML1-calmodulin signaling pathway, thereby enhancing the activity of transcription factor EB (TFEB) and promoting autophagy. Furthermore, HB alleviated pyroptosis and oxidative stress in the spinal cords of SCI mice, reducing neuronal damage. The beneficial effects of HB were reversed when the autophagy inhibitor 3MA was used. This study demonstrated that HB promoted functional recovery in SCI mice by activating TFEB through the AMPK-TRPML1-calmodulin signaling pathway, inhibiting neuroinflammation, and promoting axon regeneration. Additionally, HB demonstrated significant efficacy in alleviating ulcerative colitis (UC) in a mouse model. Mechanistic investigations revealed that HB enhanced mucosal barrier function and reduced inflammatory cytokine production by activating the AMPK signaling pathway [49].

### 5.3. Antioxidant Activity

Heart disease, particularly ischemia-reperfusion injury and myocardial infarction, is often associated with significant oxidative stress, leading to the excessive production of ROS and free radicals, which cause cellular and tissue damage [50,51]. Early studies have demonstrated that certain bioactive compounds of *P. heterophylla*, such as saponins and polysaccharides, can reduce oxidative stress and protect the myocardium [52]. For cyclic peptides from *P. heterophylla*, Xu et al. [26] isolated PA–PC from *P. heterophylla* and investigated the cardioprotective effects of these cyclic peptides on Na_2_S_2_O_4_-induced H9C2 cells hypoxia-reoxygenation (H/R) injury in vitro. The study determined that the IC_50_ of PA–PC in H9C2 cells was 14.06 μM. Furthermore, these cyclic peptides exhibited significant cardioprotective effects by counteracting oxidative stress and alleviating myocardial H/R injury. Especially in the Na_2_S_2_O_4_-induced H/R injury model, PA–PC demonstrated stronger protective effects on H9C2 cardiomyocytes compared to polydatin, a widely used natural cardioprotective agent. As new representatives of alkaloids with a β-carboline skeleton, PA–PC hold promise as potential candidates for the treatment of cardiovascular diseases by inhibiting ROS generation and improving mitochondrial function.

### 5.4. Anti-Tussive Activity

Chronic obstructive pulmonary disease (COPD) is primarily caused by smoking and is characterized by chronic airflow limitation [53]. Recent studies have identified that cyclic peptide extracts (CPEs) from *P. heterophylla* possess therapeutic efficacy for treating COPD. Yang et al. [54] administered various doses (0, 200, 400 mg/kg/day) of CPEs of *P. heterophylla* to mice with COPD of lung-qi insufficiency syndrome (LQIS-COPD) for 30 days. The treatment resulted in significant improvements in the phenotypic manifestations, such as coughing and wheezing. Lung function tests indicated that airway resistance (RL) was significantly reduced, while dynamic compliance (Cdyn) was dose-dependently increased in the treated rats. Lu et al. [55] isolated CPEs from *P. heterophylla* using a multi-step purification procedure and investigated its anti-inflammatory activity and underlying mechanisms. Rats with COPD induced by solid combustible smoke (SCS) were treated with 200, 400, and 500 mg/kg of CPEs, and samples were collected 15 days later. The results showed that CPEs significantly reduced alveolar destruction and lung inflammation, increased alveolar space, and improved lung function in COPD rats. The mechanism may involve the suppression of the TLR4-MyD88-JNK/p38 signaling pathway. The anti-tussive activity of cyclic peptides from *P. heterophylla* is shown in Table 5.

### 5.5. Hypoglycemic Activity

Studies have demonstrated that PE (0.5, 1 μM) promotes the differentiation of preadipocytes into mature adipocytes and enhances glucose uptake by adipocytes in a high-glucose (30 mM) environment under insulin stimulation. These findings suggested that PE may possess potential anti-diabetic effects by improving insulin sensitivity and accelerating glucose absorption [56]. Studies [57,58] have demonstrated that HB displayed favorable binding affinities to both DPP4 and GLP-1R, with values of −10.4 kcal/mol (for DPP4) and −9.5 kcal/mol (for GLP-1R), as reported in the respective investigations. As shown in the first study, HB’s cyclic structure was able to embed within DPP4’s catalytic cavity, establishing hydrogen bonds through stable interaction with Ser630 and Arg125, and generating π-π interactions via binding with Tyr547 and Tyr662. This mechanism inhibited DPP4 activity, thereby reducing GLP-1 degradation. Another study proposed that HB’s cyclic architecture could form hydrogen bonds with transmembrane residues of GLP-1R (such as Lys197, Tyr148, and Trp33) and π-π interactions with aromatic residues (e.g., Trp/Tyr), potentially mimicking GLP-1 to activate the receptor. Collectively, these results suggest that HB may exert dual roles: inhibiting DPP4 to preserve GLP-1 bioactivity and serving as a potential GLP-1R agonist to enhance insulin secretion, thus offering a comprehensive improvement of glycemic control through the incretin system.

### 5.6. Promoting Angiogenic Activity

Lin et al. [59] investigated the effects and mechanisms of HB on angiogenesis. The study noted that HB stimulated the growth of human umbilical vein endothelial cells (HUVECs) via the MAPK signaling pathway and subsequent downstream ERK1/2 signaling cascade. Additionally, HB stimulated cell migration and proliferation by upregulating the expression of VEGF. The angiogenic effect of HB was further validated in vivo using the chicken embryo chorioallantoic membrane (CAM) model. The promoting angiogenic activity of cyclic peptides from *P. heterophylla* is shown in Table 6.

### 5.7. Modulating Gut Microbiota

Chen et al. [49] demonstrated that the therapeutic effects of HB in ulcerative colitis (UC) are closely associated with the restoration of gut microbiota balance. In UC mice, HB regulated the gut microbiota by increasing the relative abundance of the beneficial bacterium *Akkermansia muciniphila*, thereby alleviating UC-related symptoms. Furthermore, a study on pulmonary fibrosis (PF) [60] revealed that HB may mitigate bleomycin (BLM)-induced PF by regulating the gut–lung axis. The study indicated that HB promotes the enrichment of the gut bacterium *Muribaculum intestinale* (*M. intestinale*) and its metabolite, 3-hydroxybutyric acid (3-HA), which preserve the intestinal barrier and stabilize gut immunity. In addition, HB inhibited the expression of indoleamine 2,3-dioxygenase 1 (IDO1), thereby exhibiting ferroptosis and exerting therapeutic effects through multiple mechanisms. The gut microbiota-modulating activity of cyclic peptides from *P. heterophylla* is shown in Table 7.

### 5.8. Enhancing Cognitive Function

Several studies have suggested that HB may enhance memory and could serve as a potential therapeutic agent for Alzheimer’s disease (AD). Deng et al. [61] investigated the effects of HB on AD and found that HB improved memory by promoting neurite regeneration and regulating neuroinflammation in mouse models induced by amyloid-beta (Aβ). In particular, HB protected neurons from apoptosis and axonal atrophy induced by Aβ25-35. Furthermore, HB increased the activity of T-helper cells in the spleen, reduced neuroinflammation, and enhanced cognitive functions, including memory retrieval and spatial memory. Network pharmacology analysis identified the key molecular targets of HB neuroprotectors, including MMP2, MMP9, and Src, which were significantly upregulated after coculturing with HB (10 μg/mL) for 4 days. In vivo studies demonstrated that HB treatment could increase the density of synaptic proteins associated with cognitive enhancement in the mouse brain and ultimately improve memory. Feng et al. [62] employed both the zebrafish model and the APP/PS1 transgenic mouse model to validate the neuroprotective and memory-enhancing effects of *P. heterophylla* extract. Their findings showed that HB could promote neuron survival, stimulate neurite outgrowth, and regulate the expression of apoptosis-related genes such as P53 and Caspase 3.

Yang et al. [63] discovered that HB could penetrate the blood–brain barrier (BBB) and enter the central nervous system. First, it promoted neuronal neurite outgrowth to construct denser neural networks and simultaneously upregulated the expression of synaptic proteins to strengthen synaptic connections. Second, it optimized the efficiency of neurotransmission by regulating the metabolic balance of monoamine neurotransmitters such as 5-hydroxytryptamine and dopamine, ultimately synergistically enhancing cognitive function. In this study, ICR mice were treated with an oral gavage of HB at 10 g/kg, and plasma and cerebral cortex tissues were collected 30 min later. After methanol extraction, the samples were analyzed by ultra-high performance liquid chromatography–tandem mass spectrometry (UHPLC-MS/MS). The presence of HB in biological samples was confirmed by comparing the accurate mass (error ± 0.5 mmu) of its quasi-molecular ion peak (M^+^H^+^ = 779.4435) and MS/MS fragment ions (e.g., *m/z* 654.3957, 569.3074, etc.) with the extracted ion chromatogram and fragmentation pattern of the standard. The qualitative results showed that characteristic chromatographic peaks of HB (retention time: approximately 12.6–12.7 min) were detected in both the plasma and cerebral cortex at 30 min after oral administration, while no such component was detected in the blank control group, confirming that HB could be absorbed through the oral route and penetrate the BBB. The quantitative results showed that the concentration of HET-B in the cerebral cortex was 414.4 ± 91.1 ng/g, indicating that it reached a biologically active dose level in the brain. The enhancing activity of cyclic peptides from *P. heterophylla* on cognitive function is summarized in Table 8.

### 5.9. Inhibiting Tyrosinase Activity

Tyrosinase is a key enzyme present in both animals and plants, playing a critical role in melanin synthesis and the aging process in humans. Li et al. [64] extracted tyrosinase from potatoes and assessed the inhibitory effect of HB on tyrosinase activity using UV–Visible spectrophotometry. The results indicated that at a concentration of 4.98 μg/mL, HB reduced tyrosinase activity by 43%. Morita et al. [19,20,21,65] isolated cyclic peptides PA–PH from *P. heterophylla* and evaluated their tyrosinase inhibitory activities using the dopachrome method. The findings revealed that the IC_50_ values for the peptides ranged from 50 μM to 187 μM, with PF showing the strongest inhibition (IC_50_ = 50 μM), followed in order by PC (IC_50_ = 63 μM), PG (IC_50_ = 75 μM), PD (IC_50_ = 100 μM), PA (IC_50_ = 131 μM), PE (IC_50_ = 175 μM), and PB (IC_50_ = 187 μM). The inhibitory activity of PH was relatively weak, with only 15% inhibition at a concentration of 800 μM. Additionally, the researchers investigated the inhibition of melanogenesis in melanoma cells, finding the following IC_50_ values: PA (IC_50_ = 2.5 μM), PD (IC_50_ = 49 μM), PG (IC_50_ = 102 μM), and PC (IC_50_ = 171 μM). The inhibiting activity of cyclic peptides from *P. heterophylla* on tyrosinase is shown in Table 9.

### 5.10. Anti-Fibrotic Effects

Studies have shown that HB may alleviate pulmonary fibrosis (PF) by regulating the gut microbiota and ferroptosis, inhibiting the transdifferentiation of lung fibroblasts, and other mechanisms. In terms of regulating the gut–lung axis of bleomycin-induced PF mice, Chen et al. [60] found that HB could restore intestinal barrier function by enriching the gut microbiota (e.g., *M. intestinale*) and its metabolite, 3-HA. This restoration enhanced intestinal barrier function and inhibited Indoleamine 2, 3-dioxygenase 1 (IDO1)-mediated ferroptosis, and it suppressed TGF-β1-induced epithelial-to-mesenchymal transition (EMT) in both in vivo and in vitro models, thereby improving PF. Elevated IDO1 expression and ferroptosis were observed in the lung tissues of patients with idiopathic PF, further confirming the critical role of IDO1 in PF progression. This research identified new therapeutic targets and strategies for PF treatment. Shi et al. [66] investigated the protective effects of HB on BLM-induced PF in mice, focusing on the underlying mechanism of fibrosis. The PF mice exhibited typical symptoms such as alveolar collapse, inflammatory cell infiltration, and collagen deposition. Treatment with HB alleviated these symptoms, improving the body weight, survival rate, and lung health in the mice. Mechanistic analyses showed that HB activates AMPK, which in turn inhibits the expression of STING and reduces the levels of fibrosis markers, including COL-1 and α-SMA. Additionally, HB significantly suppressed EMT and the transdifferentiation of lung fibroblasts, further supporting its potential therapeutic role in the treatment of PF through modulation of the TGF-β1-Smad2/3 and AMPK-STING signaling pathways. The anti-fibrotic effects of cyclic peptides from *P. heterophylla* are shown in Table 10.

## 6. Conclusions and Prospectives

With thousands of years of clinical experience, TCM has provided valuable insights for the development of natural bioactive substances, including linear and cyclic peptides. For instance, TCM-isolated peptides, such as soybean peptides, have been shown to possess significant antihypertensive effects [67]. Additionally, two peptides from a folk Chinese medicine, *Blaps rhynchopetera*, have been reported to exhibit excellent antifungal activity [68,69]. Unlike linear peptides, the cyclic structure of *P. heterophylla* peptides might lead to better biological activities. Cyclic peptides from *P. heterophylla* have demonstrated promising pharmacological activities both in vitro and in vivo. Research on these peptides has primarily focused on their activities in inflammation, oxidative stress, cancer, chronic obstructive pulmonary disease, diabetes, Alzheimer’s disease, pulmonary fibrosis, and other diseases. The underlying mechanisms might involve the regulation of cell proliferation, migration, and cell death, as well as the modulation of inflammatory cytokine levels, blood glucose levels, and gut microbiota. Additionally, these peptides might promote neurite regeneration and inhibit epithelial–mesenchymal transition.

Although cyclic peptides from *P. heterophylla* have been reported to exhibit various bioactivities, the research on their structure–activity relationship (SAR) remains undeveloped, which limits studies of the medicinal potential of *P. heterophylla* cyclic peptides. To investigate the SAR of cyclic peptides from *P. heterophylla*, it is necessary to synthesize a series of *P. heterophylla* cyclic peptides and their analogues by the SPPS method, and their biological activities can be determined to analyze the influence of the types and positions of amino acid residues on the biological effects, which will benefit the improvement of the activities of cyclic peptides from *P. heterophylla*.

Moreover, the unique structural features of cyclic peptides from *P. heterophylla* contribute to their stability, which makes them viable candidates as carriers for targeted drug delivery. It is possible to develop targeted drug delivery systems based on cyclic peptides from *P. heterophylla* by integrating nanotechnology, which could improve their targeting ability and bioavailability, thereby enhancing therapeutic efficacy, especially in cancer and immune diseases. However, challenges such as the formulation of these delivery systems could often involve complicated processes to ensure the proper encapsulation and release of drugs. Additionally, immunogenicity remains a concern as the introduction of novel peptide-based carriers could trigger adverse immune responses in patients. To further enhance the efficacy of cyclic peptides from *P. heterophylla*, combination therapies, such as those involving immune checkpoint inhibitors, could also be explored for synergistic anticancer effects.

In conclusion, cyclic peptides from *P. heterophylla* exhibited substantial developmental promise with structural stabilities and diverse pharmacological activities. Further in-depth studies are required to fully clarify their pharmacological activities and facilitate their effective translation into clinical applications.

## Figures and Tables

**Figure 1 molecules-30-02521-f001:**
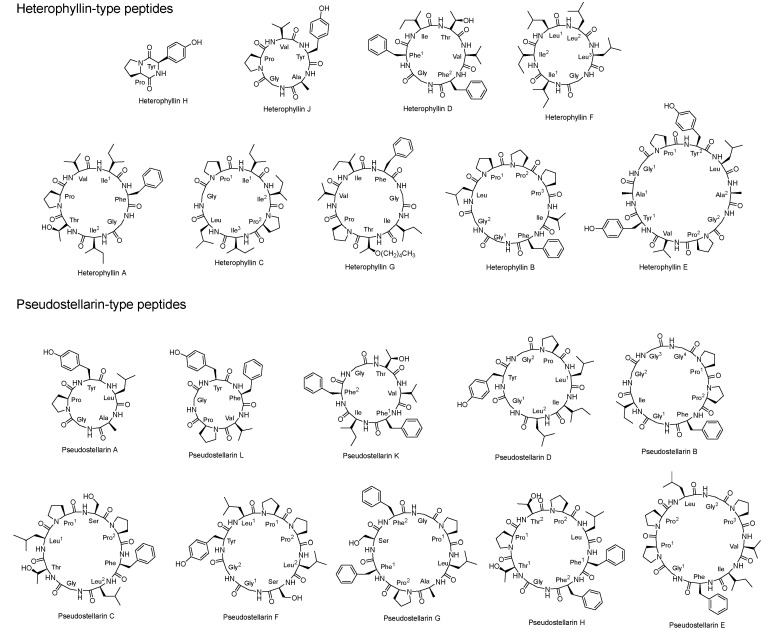
Chemical structures of cyclic peptides from *P. heterophylla*.

**Table 2 molecules-30-02521-t002:** Pharmacokinetic parameters of HB in SD rats.

Parameters	2.08 mg/kg [39]	4.16 mg/kg [39]	8.32 mg/kg [39]	20 mg/kg [40]
*C*_max_ (ng/mL)	1372 ± 148	2446 ± 242	4666 ± 192	4082.76 ± 892.20
*T*_max_ (min)	5	5	10	3.0 ± 1.8
*T*_1/2_ (min)	39.9 ± 14.6	48.6 ± 11.2	64.8 ± 14.5	574.2 ± 160.2
MRT_0–t_ (min)	23.1 ± 3.5 21	23.3 ± 3.7	21.8 ± 2.2	54.0 ± 16.2
MRT_0–∞_ (min)	26.5 ± 3.6	24.6 ± 4.3	22.6 ± 2.6	92.4 ± 25.8
AUC_0–t_ (ng·min/mL)	29,043 ± 5221	48,312 ± 2747	116,838 ± 12,018	42,100 ± 1170
AUC_0–∞_ (ng·min/mL)	29,451 ± 5033	48,489 ± 2811	117,006 ± 11,947	718.02 ± 19.77
*V*_d_ (L/kg)	4.38 ± 2.35	5.99 ± 1.18	6.79 ± 1.94	383,460.70 ± 1003.09
*CL* (L/min/kg)	0.072 ± 0.013	0.086 ± 0.005	0.071 ± 0.007	464.48 ± 12.99
*K*e (1/h)				0.08 ± 0.02

**Table 3 molecules-30-02521-t003:** Anti-tumor activity of cyclic peptides from *P. heterophylla*. (Increase, ↑; Decrease, ↓).

Compound	Types	Testing Subjects	Doses/Duration/Route	Effects	Mechanism	Ref.
HB	In vitro	OVCAR-8 andSKOV3 cells	0, 25, 50, 75, 100 μM for 24 h	Cell proliferation ↓; Colony formation↓; Apoptosis ↑;	NRF2 and HO-1 expression ↓; Involved in apoptosis induction.	[43]
		MKN-4 and BGC-823 cells	0, 10, 25, and 50 μM for 24–48 h	Cell proliferation ↓; Apoptosis ↑; Arrested cell cycle at G0/G1 phase;	Activates ER stress by upregulating IRE1, CHOP, GRP78, downregulating Bcl-2, and facilitating caspase-3 expression.	[44]
		ECA-109 cells	10, 25, 50 μM, 24 h	Adhesion and invasion ↓; Cell proliferation ↓; E-cadherin expression ↑;	PI3K/AKT/β-catenin pathway ↓; Snail, Vimentin, MMP-2/9 ↓; E-cadherin ↑;	[45]
		HGC-27 and AGS cells	0, 10, 20, 40, 60, 80, 100 μM for 24 h	Cell viability, colony formation, migration, and invasion ↓;	Binds to CXCR4, PI3K/AKT signaling pathway; PD-L1 expression, metastasis ↓;	[46]
	In vivo	OVCAR8 xenografted nude mice	20 mg/kg/day, i.p., 35 days	Tumor growth ↓; Tumor volume ↓; Ki67 expression ↑;	Tumor proliferation via NRF2/HO-1 inhibition; Apoptosis in tumor tissues ↑;	[43]
		MKN-45 cells-induced tumor model in nude mice	10, 15 mg/kg/2 days, i.p., 28 days	Tumor growth, tumor weight ↓; IRE1, CHOP, GRP78 ↑; Bcl-2 ↓;	ER stress activation through IRE1, CHOP, GRP78, Bcl-2 suppression, leading to apoptosis and inhibited tumor growth.	[44]
		HGC-27 cells-induced tumor metastasis model in nude mice	500 mg/kg/day, i.g., 14 days	Tumor metastasis ↓; Lung metastatic nodules ↓;	Targets CXCR4 and modulates PI3K/AKT signaling; PD-L1 expression, tumor cell migration/invasion ↓;	[46]

**Table 4 molecules-30-02521-t004:** Anti-inflammatory and antioxidant activity of cyclic peptides from *P. heterophylla*. (Increase, ↑; Decrease, ↓).

Compound	Types	Testing Subjects	Doses/Duration/Route	Effects	Mechanism	Ref.
HB	In vitro	LPS-stimulated RAW 264.7 cells	25, 50, 100 μM, 1 h	LPS-induced NO, IL-6, and IL-1β production ↓; ROS generation ↓; Apoptosis ↓;	Suppression of inflammation and oxidative stress through the PI3K/Akt pathway; p-AKT/AKT and p-PI3K/PI3K ratios ↑;	[47]
	In vivo	SCI contusion model in C57BL/6J mice	20 mg/kg/day, i.p., 3 days	Motor function ↑; Axonal regeneration ↑; Bladder recovery ↑;	Activates autophagy by enhancing TFEB translocation; Suppresses oxidative stress and pyroptosis by the AMPK-TRPML1-calcineurin pathway.	[48]
		DSS (4% Dextran Sulfate Sodium)-induced colitis in C57BL/6J mice	20, 80 mg/kg/day, i.g., 7 days	Disease activity index↓; Colon length ↑; Histological damage, and inflammation ↓;	Restored the intestinal mucosal barrier, improved microbiota composition, and reduced inflammatory cytokines.	[49]

**Table 5 molecules-30-02521-t005:** Anti-tussive activity of cyclic peptides from *P. heterophylla*. (Increase, ↑; Decrease, ↓).

Extract	Types	Testing Subjects	Doses/Duration/Route	Effects	Mechanism	Ref.
CPE	In vitro	Alveolar macrophages	50, 100, 200, 500, 1000, and 2000 μg/mL for 12, 24, and 48 h	TNF-α release ↓; IL-10 release ↑; Inflammation ↓;	TLR4, MyD88, and AP-1 mRNA levels ↓; JNK/p38 signaling pathways ↓; Pro-inflammatory cytokine release ↑;	[55]
	In vivo	Smoky environment + Papain aerosol to induce LQIS-COPD rats	100, 200, 400 mg/kg/d, i.g., 30 days	Cough, shortness of breath, wheezing ↓; RL ↓; Balanced Cdyn;	Improved lung function by reducing R_L_ and enhancing Cdyn; effects may be due to the restoration of lung qi and inhibition of inflammation.	[54]
		SCS-induced COPD model rats	200, 400, 500 mg/kg/day, p.o., 15 days	Lung function ↑; Alveolar destruction ↓; Alveolar space ↑;	TLR4-MyD88 signaling pathway ↓; Inflammatory cytokines (TNF-α, IL-10) ↓; Lung tissue morphology and function ↑;	[55]

**Table 6 molecules-30-02521-t006:** Promoting the angiogenic activity of cyclic peptides from *P. heterophylla*. (Increase, ↑; Decrease, ↓).

Compound	Types	Testing Subjects	Doses/Duration/Route	Effects	Mechanism	Ref.
HB	In vitro	HUVEC cells	0–200 μg/mL for 48 h	Cell proliferation ↑ (0–100 μg/mL HB); Cell proliferation ↓ (150–200 μg/mL HB);	Promoted cell proliferation through increased VEGF expression and activation of the MAPK signaling pathway (Ras/Raf/Mek/Erk).	[58]
HB	In vivo	CAM	0–10 mg/mL/day, 3 days	Vascular proliferation (5 mg/mL/day) ↓; Blood vessel growth (10 mg/mL/da) ↑;	Promoted angiogenesis by influencing growth factors (e.g., VEGF) and their signaling pathways.	[58]

**Table 7 molecules-30-02521-t007:** Modulating gut microbiota activity of cyclic peptides from *P. heterophylla*. (Increase, ↑; Decrease, ↓).

Compound	Types	Testing Subjects	Doses/Duration/Route	Effects	Mechanism	Ref.
HB	In vitro	NCM460 cells	0.1–10 μM, 24 h	Occludin and ZO-1 expression ↑; Disruption of the epithelial barrier induced by TNF-α ↓;	AMPK signaling ↑; Maintain intestinal epithelial barrier function.	[49]
HB	In vivo	DSS-induced colitis in mice.	20, 80 mg/kg/day, i.g., 7 days	Colon length ↑; Inflammation ↓; Deterioration of the intestinal mucosal barrier.	Protects beneficial intestinal bacteria, restores gut health, and protects against colitis.	[49]

**Table 8 molecules-30-02521-t008:** The enhancing activity of cyclic peptides from *P. heterophylla*. on cognitive function (Increase, ↑; Decrease, ↓).

Compound	Types	Testing Subjects	Doses/Duration/Route	Effects	Mechanism	Ref.
HB	In vitro	Aβ25-35-induced primary cortical neurons	1, 10 μM, 4 days	Neuron death ↓;	Apoptosis markers (Bax) ↓; Anti-apoptotic markers (Bcl-2) ↑; Synaptic protein levels were preserved;	[61]
		SH-SY5Y cells	0.1–10 μM, 1–3 days	Aβ-induced cell damage ↓;	Aβ-induced neuronal apoptosis ↓; Synaptic regeneration↑;	[61]
		Aβ25-35-induced primary cortical neurons	1–100 μM	Neuronal survival rate ↑; Aβ25-35-induced cell damage ↓;	Expression of apoptosis-related genes (P53 and Caspase3) ↓; β3-tubulin and MAP2-positive neurite density ↑; Synaptic plasticity ↑;	[62]
		Primary cortical neurons	0.011 μM, 4 days	β3-tubulin-positive neurite density ↑; Neurite outgrowth ↑;	Promotes cognitive enhancement and neuroprotection through neuronal growth.	[63]
	In vivo	Aβ1-42 i.c.v.-induced AD mice	1, 10 μM/kg/day, i.p., 16 days	Memory deficits ↓; Cognitive performance ↑; Neuroinflammation. ↓;	Microglial activation ↓; Axonal regeneration ↑; Modulation of inflammatory cytokines.	[61]
		APP/PS1 transgenic mice	10 μM/kg/day, i.p., 60 days	Novel object recognition and spatial memory ↑; Exploration of new locations;	Expression of apoptosis-related genes (P53 and Caspase3) ↓; Neuronal survival rate, neurite density, neural connectivity ↑;	[62]
		ICR mice	1 and 10 μM/kg, i.p., 10 days	Object recognition and location memory ↑; Neurite outgrowth ↑;	Penetrates the blood–brain barrier and regulates dopamine turnover, leading to cognitive improvements and enhanced memory function.	[63]

**Table 9 molecules-30-02521-t009:** Inhibiting tyrosinase activity of cyclic peptides from *P. heterophylla*.

Compounds	Type	Testing Subjects	Effects	Ref.
PA	In vitro	Tyrosinase solution	IC_50_: 187 μM	[20]
PB		Tyrosinase solution	43% tyrosinase inhibition at 4.98 μg/mL	[65]
PB		Tyrosinase solution	IC_50_: 187 μM	[20]
PC		Tyrosinase solution	IC_50_: 63 μM	[20]
PC		Mouse B16 melanoma cells	IC_50_: 134 μM	[20]
PD		Tyrosinase solution	IC_50_: 100 μM	[21]
PD		Mouse B16 melanoma cells	IC_50_: 49 μM	[21]
PE		Tyrosinase solution	IC_50_: 175 μM	[21]
PF		Tyrosinase solution	IC_50_: 50 μM	[21]
PG		Tyrosinase solution	IC_50_: 75 μM	[3]
PH		Tyrosinase solution	15% tyrosinase inhibition at 800 μM	[66]

**Table 10 molecules-30-02521-t010:** Anti-fibrotic effects of cyclic peptides from *P. heterophylla* (Increase, ↑; Decrease, ↓).

Compound	Types	Testing Subjects	Doses/Duration/Route	Effects	Mechanism	Ref.
HB	In vitro	MLE-12 cells	0.5 mg/mL, 1 day	ROS ↓; Ferroptosis ↓; Preserved mitochondrial mass;	Suppressed ferroptosis via IDO1-mediated Fe^2+^ accumulation and oxidative stress reduction.	[60]
		A549 cells	0.5 mg/mL, 1 day	Lung fibrosis ↓; ROS and collagen deposition ↓;	IDO1-mediated ferroptosis suppression.	[60]
		NCM460 cells	0.5 mg/mL, 1 day	Restored colonic epithelial integrity; Oxidative damage ↓;	Modulated ferroptosis and re-established intestinal mucosal barrier.	[60]
		MLE-12 cells	1, 10, 20, 50, 100 μM, 1 day	Inhibited TGF-β1-induced EMT in alveolar cells;	Inhibited the expression of Vimentin and promoted E-cadherin expression; AMPK activation reduced STING expression.	[66]
		Primary lung fibroblasts	1, 10, 20, 50, 100 μM, 2 days	Fibroblast transdifferentiation ↓; ECM deposition ↓;	AMPK pathway ↑; STING expression ↓; TGF-β1 signaling in fibroblasts ↓;	[66]
	In vivo	BLM-induced PF in mice	20 mg/kg/day, i.g., 21 days	Collagen I deposition ↓; Restored intestinal barrier;	Enriched *M. intestinale* and promoted the production of 3-HA, which modulates IDO1-mediated ferroptosis.	[60]
		Antibiotic-induced microbiota depletion in mice	20 mg/kg/day, i.g., 7 days	Restored intestinal mucosal barrier; PF symptoms↓;	Increased 3-HA production and reprogramed the intestinal ecosystem to modulate the gut microbiota and alleviate PF.	[60]
		C57BL/6 Mice (PF model induced by BLM):	5 and 20 mg/kg/day, p.o., 14 days	Fibrosis and collagen deposition in lungs ↓;	AMPK ↑; STING expression ↓; TGF-β1/Smad2/3 signaling ↓; ECM deposition and fibroblast accumulation ↓.	[66]

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
