# Peer review of "Chemical Properties, Preparation, and Pharmaceutical Effects of Cyclic Peptides from Pseudostellaria heterophylla"

_molecules, 2025, doi:10.3390/molecules30122521_

Round 1

Reviewer 1 Report

Comments and Suggestions for Authors

Chemical Property, Preparation, and Pharmaceutical Effects of 2 Cyclic Peptides from Pseudostellaria heterophylla  with authors : Yue Yang, Luan Wen, Zhuang-Zhuang Jiang, Ben Chung-Lap Chan, Ping-Chung Leung, Chun-Kwok Wong and Ning‑Hua Tan

With this review, the authors provide in-depth information on a topical topic, such as the use of cyclic peptides. Cyclic peptides, especially those derived from natural products, are considered an important source of drugs and more than 40 cyclic peptides have been approved for clinical use. This review presents information on cyclic peptides from P. heterophylla, describing their biological activities and potential mechanisms.

The chemical characterization of 19 cyclic peptides from P. heterophylla, as well as their chemical structures, are presented in depth.

An important part of the review is the description of the preparation methods - mainly by direct extraction, biosynthesis and chemical synthesis.

Much of the review is devoted to the interest in these peptides, explained by their pharmacokinetic characteristics. Their pharmacological activity is well described, including antitumor, anti-inflammatory, antioxidant, immunomodulatory, memory-enhancing, etc., which highlights their potential for application and development.

Cyclic peptides with antitussive, hypoglycemic, angiogenic activity, modulation of intestinal microbiota, improvement of cognitive function, inhibition of tyrosinase activity, antifibrotic effects, etc. are also described in detail.

The presented work provides detailed information that can be used for future research and development of cyclic peptides from P. heterophylla as potential therapeutic agents.

Since the pharmacological activity of cyclic peptides from P. heterophylla is not yet fully understood, this provides an opportunity to direct research for suitable candidates for carriers for targeted drug delivery to develop targeted drug delivery systems based on cyclic peptides.

Cyclic peptides are also involved in cell regeneration and are used for tissue repair. It would be nice to add information on this topic.

After corrections and additions, the article can be published.

Author Response

Reviewer 1

With this review, the authors provide in-depth information on a topical topic, such as the use of cyclic peptides. Cyclic peptides, especially those derived from natural products, are considered an important source of drugs and more than 40 cyclic peptides have been approved for clinical use. This review presents information on cyclic peptides from P. heterophylla, describing their biological activities and potential mechanisms.

The chemical characterization of 19 cyclic peptides from P. heterophylla, as well as their chemical structures, are presented in depth.

An important part of the review is the description of the preparation methods - mainly by direct extraction, biosynthesis and chemical synthesis.

Much of the review is devoted to the interest in these peptides, explained by their pharmacokinetic characteristics. Their pharmacological activity is well described, including antitumor, anti-inflammatory, antioxidant, immunomodulatory, memory-enhancing, etc., which highlights their potential for application and development.

Cyclic peptides with antitussive, hypoglycemic, angiogenic activity, modulation of intestinal microbiota, improvement of cognitive function, inhibition of tyrosinase activity, antifibrotic effects, etc. are also described in detail.

The presented work provides detailed information that can be used for future research and development of cyclic peptides from P. heterophylla as potential therapeutic agents.

Since the pharmacological activity of cyclic peptides from P. heterophylla is not yet fully understood, this provides an opportunity to direct research for suitable candidates for carriers for targeted drug delivery to develop targeted drug delivery systems based on cyclic peptides.

Cyclic peptides are also involved in cell regeneration and are used for tissue repair. It would be nice to add information on this topic.

After corrections and additions, the article can be published.

Responses Thank you very much for your supportive and valuable comments on our review article. We have corrected and revised our manuscript accordingly.

Reviewer 2 Report

Comments and Suggestions for Authors

This manuscript, “Chemical Property, Preparation, and Pharmaceutical Effects of Cyclic Peptides from Pseudostellaria heterophylla” by Chan and co-workers presents a comprehensive review of the chemistry, extraction, synthesis, pharmacokinetics, and pharmacological activities of cyclic peptides derived from Pseudostellaria heterophylla. The topic is timely and relevant, given the growing interest in plant-derived cyclic peptides for therapeutic applications. The manuscript is well-organized, and the extensive literature coverage reflects a significant effort. However, several areas require clarification and refinement to meet the standards of Molecules:

1) Line 19-20, “promoting fluid in body” should be “promoting body fluid production”

2) Line 22, abstract, the sentence “These cyclic peptides feature distinct cyclic structures…” is redundant.

3) Also, adding specific biological effects or standout pharmacological potentials like antitumor, cognitive benefits, etc. can improve the weight of abstract.

4) To underline the novelty of this review, the authors should mention in the introduction, how P. heterophylla cyclic peptides differ from other plant-derived peptides like cyclotides, orbitides etc. Additionally, addressing limitations or gaps in current scientific understanding is necessary.

5) In Table 1, include physicochemical parameters like logP, H-bond donors/acceptors, if available.

6) Figure 1 in the current resolution and layout, is visually compressed and lacks clarity. Consider reorganizing the figure to increase the size of the peptide structures.

7) In section 2, the authors might briefly discuss whether any SARs have been observed across different ring sizes or amino acid compositions. For instance, are hydrophobic or Pro-rich motifs associated with distinct bioactivities? Even a brief commentary, grounded in the cited literature, would be informative.

8) The section 3 reads more like a listing than a comparative discussion. It would be helpful to include a short analysis comparing the efficiency, scalability, and practical challenges of extraction, biosynthesis, and chemical synthesis approaches.

9) The pharmacokinetics section is too concise given the depth of data reported. Including a summary table of key parameters (Cmax, Tmax, T1/2, Vd, CL) for HB and other peptides would aid clarity. The discussion could be expanded to address the significance of the dual membrane-penetration mechanisms proposed for HB.

10) In antitumor section, some mechanistic details are well presented (e.g., NRF2/HO-1, ER stress), while others (e.g., CXCR4/PI3K/PD-L1) require more molecular-level interpretation. It would be useful to clarify whether these mechanisms act independently or synergistically.

11) In anti-inflammatory and antioxidant  section, the explanation of HB’s role in spinal cord injury and colitis is detailed. Consider integrating more mechanistic comparisons across models. For instance, is AMPK a shared mediator in both effects?

12) The mechanistic depth varies across the biological activities. For example, the explanation of HB's DPP4/GLP-1 interactions is superficial. Molecular docking should be supported with scoring data, if available.

13) The claim that HB penetrates the blood–brain barrier within 30 minutes is noteworthy. Authors should elaborate on the experimental methods (LC-MS protocol, quantification strategy) and the validation supporting this observation.

14) In conclusion, “significant value in clinical applications” should be softened to reflect their current preclinical status. The suggestion to develop targeted drug delivery systems is a promising direction, but mentioning practical challenges involved, such as formulation complexity, immunogenicity, or in vivo stability is necessary.

Comments on the Quality of English Language

The manuscript is generally readable and conveys the scientific content adequately. However, there are multiple instances of awkward phrasing, grammatical errors, and inconsistent article usage throughout the text.

Author Response

Comments and Suggestions for Authors

This manuscript, “Chemical Property, Preparation, and Pharmaceutical Effects of Cyclic Peptides from Pseudostellaria heterophylla” by Chan and co-workers presents a comprehensive review of the chemistry, extraction, synthesis, pharmacokinetics, and pharmacological activities of cyclic peptides derived from Pseudostellaria heterophylla. The topic is timely and relevant, given the growing interest in plant-derived cyclic peptides for therapeutic applications. The manuscript is well-organized, and the extensive literature coverage reflects a significant effort. However, several areas require clarification and refinement to meet the standards of Molecules:

  • Line 19-20, “promoting fluid in body” should be “promoting body fluid production”

Response: Thank you for your comments, we have rewritten the sentence accordingly.

  • Line 22, abstract, the sentence “These cyclic peptides feature distinct cyclic structures…” is redundant.

Response: Thank you for your comments, we have deleted the phrases accordingly.

  • Also, adding specific biological effects or standout pharmacological potentials like antitumor, cognitive benefits, etc. can improve the weight of abstract.

Response: Thank you for your comments. We have added cognitive benefits to the biological effect description.

4) To underline the novelty of this review, the authors should mention in the introduction, how P. heterophylla cyclic peptides differ from other plant-derived peptides like cyclotides, orbitides etc. Additionally, addressing limitations or gaps in current scientific understanding is necessary.

Response: Thank you for your comments. We have added the differences between P. heterophylla cyclic peptides and other cyclic peptides in the introduction.

5) In Table 1, include physicochemical parameters like logP, H-bond donors/acceptors, if available.

Response: Thank you very much for your suggestion. However, the studies of physicochemical parameters of other cyclic peptides from P. heterophylla are limited. In this way, we did not add a new column in Table 1.

6) Figure 1 in the current resolution and layout, is visually compressed and lacks clarity. Consider reorganizing the figure to increase the size of the peptide structures.

Response: Thank you for your comments. We have reorganized the figure according to the structural types and the size of the peptide structures, and re-exported it.

7) In section 2, the authors might briefly discuss whether any SARs have been observed across different ring sizes or amino acid compositions. For instance, are hydrophobic or Pro-rich motifs associated with distinct bioactivities? Even a brief commentary, grounded in the cited literature, would be informative.

Response: Thank you very much for your suggestion. We have discussed the potential activity difference across different ring sizes of cyclic peptides, though the SAR studies of cyclic peptides from P.heterophylla are limited.

8) The section 3 reads more like a listing than a comparative discussion. It would be helpful to include a short analysis comparing the efficiency, scalability, and practical challenges of extraction, biosynthesis, and chemical synthesis approaches.

Response: Thank you very much for your kind comments. We have rewritten Section 3 according to your suggestion. We reorganized Section 3.1 into two parts: extraction and purification. As for Sections 3.2 and 3.3, we added the overall yields to describe the synthesis efficiency.

9) The pharmacokinetics section is too concise given the depth of data reported. Including a summary table of key parameters (Cmax, Tmax, T1/2, Vd, CL) for HB and other peptides would aid clarity. The discussion could be expanded to address the significance of the dual membrane-penetration mechanisms proposed for HB.

Response: Thank you for the comments. Following your suggestions, we have supplemented the content related to pharmacokinetics and presented the relevant parameters in table 2.

10) In antitumor section, some mechanistic details are well presented (e.g., NRF2/HO-1, ER stress), while others (e.g., CXCR4/PI3K/PD-L1) require more molecular-level interpretation. It would be useful to clarify whether these mechanisms act independently or synergistically.

Response: Thank you for the comments. We have added the CXCR4/PI3K/PD-L1 related mechanisms.

11) In anti-inflammatory and antioxidant section, the explanation of HB’s role in spinal cord injury and colitis is detailed. Consider integrating more mechanistic comparisons across models. For instance, is AMPK a shared mediator in both effects?

Responses: Thank you for the comments. We have added the role of HB in spinal cord injury and enteritis, and the associated mechanisms, and discussed the co-role of AMPK in both models.

12) The mechanistic depth varies across the biological activities. For example, the explanation of HB's DPP4/GLP-1 interactions is superficial. Molecular docking should be supported with scoring data, if available.

Responses: Thank you for the comments. We have expanded the interpretation of the HB DPP4/GLP-1 interaction and provided more data on molecular docking.

13) The claim that HB penetrates the blood–brain barrier within 30 minutes is noteworthy. Authors should elaborate on the experimental methods (LC-MS protocol, quantification strategy) and the validation supporting this observation.

Responses: Thank you very much for the positive comments. Following your suggestions, we revised and expanded the relevant content and reorganized the structure of the paragraphs.

14) In conclusion, “significant value in clinical applications” should be softened to reflect their current preclinical status. The suggestion to develop targeted drug delivery systems is a promising direction, but mentioning practical challenges involved, such as formulation complexity, immunogenicity, or in vivo stability is necessary.

Responses: Thank you very much for the positive comments. We have modified the expression for clinical applications and supplemented the challenges that may be faced in building a targeted drug delivery system

15) Comments on the Quality of English Language

The manuscript is generally readable and conveys the scientific content adequately. However, there are multiple instances of awkward phrasing, grammatical errors, and inconsistent article usage throughout the text.

Response: Thank you very much for your comments. We have made our best effort to improve the English throughout the entire manuscript.

Reviewer 3 Report

Comments and Suggestions for Authors

The Manuscript entitled: Chemical Property, Preparation, and Pharmaceutical Effects of Cyclic Peptides from Pseudostellaria heterophylla, could be considered for acceptation after English editing and other minor revisions.

Introduction (Lines 34–38)

The sentence in this section needs to be rewritten. The current wording suggests that the terminology of Traditional Chinese Medicines (TCMs) may not be familiar to some researchers and medical professionals. It would be helpful for the authors to provide brief explanations or draw parallels with terminology used in conventional Western medicine.

Figure 1

It would enhance the clarity and informativeness of Figure 1 if the authors included representative structural types, such as heterophyllin-type and pseudostellarin-type peptides.

Lines 81–82

The authors are requested to elaborate on the following statement: “However, recent studies have increasingly shown that these roots also possess significant potential for various applications.” Please specify the nature of these applications or provide relevant references.

Table 1

It would be valuable to include the method(s) of compound identification used for each of the cyclic peptides (CPs) listed in the table.

Section 3.1 (Lines 106–116)

The authors are encouraged to provide additional details regarding the peptide isolation procedures. Specifically, did the authors employ column chromatography (CC), high-performance liquid chromatography (HPLC), or another technique? Note that the term elution is commonly associated with chromatography and is not typically used in extraction methodology; please revise the related sentences accordingly.
In my professional experience, high-speed counter-current chromatography (HSCCC) is a productive method for the preliminary separation of natural products, but it often requires a follow-up purification step using HPLC. The authors should consider including the following statement in the discussion section only if it is directly relevant to the isolation of CPs:

“High-speed countercurrent chromatography (HSCCC) has been widely applied in the separation and purification of natural products in recent years due to its advantages, including the absence of irreversible adsorption, high recovery rate, and ease of operation [26]. Han et al. [27] employed…”

Section 4.

The authors should clearly state which model was employed and provide additional methodological details about the study described by Zhao et al. [41] 

Section 6.

The conclusion (lines 417–424) is intriguing and presents an interesting perspective. It would be beneficial if the authors could discuss the relationship between structure–activity data and TCM-based knowledge of this herbal drug. Additionally, this part of the conclusion would be strengthened by the inclusion of references supporting this viewpoint, possibly involving similar peptides.

Comments on the Quality of English Language

There is an apparent inconsistency in the quality of English throughout the manuscript. It appears that the conclusions may have been further edited using language-enhancement tools, which likely contributes to their clarity. The authors are advised to ensure consistency across the entire text, paying particular attention to technical terminology, especially in the sections dealing with chemistry.

Author Response

Comments and Suggestions for Authors

The Manuscript entitled: Chemical Property, Preparation, and Pharmaceutical Effects of Cyclic Peptides from Pseudostellaria heterophylla, could be considered for acceptance after English editing and other minor revisions.

1) Introduction (Lines 34–38)

The sentence in this section needs to be rewritten. The current wording suggests that the terminology of Traditional Chinese Medicine (TCM) may not be familiar to some researchers and medical professionals. It would be helpful for the authors to provide brief explanations or draw parallels with terminology used in conventional Western medicine.

Response: Thank you for your comments. We have explained for "According to the theory of TCMs, it has a slightly bitter flavor, a mild nature, and is associated with the spleen and lung meridians. It has the effect of enhancing qi, invigorating spleen, generating fluid in the body, and moistening the lung, commonly used for spleen deficiency, physical fatigue, dry cough and anorexia." and placed the explanatory sentence at the end of this paragraph.

2) Figure 1

It would enhance the clarity and informativeness of Figure 1 if the authors included representative structural types, such as heterophyllin-type and pseudostellarin-type peptides.

Response: Thank you for your comments. We have reorganized the figure according to the structural types and the size of the peptide structures, and re-exported it.

3) Lines 81–82

The authors are requested to elaborate on the following statement: “However, recent studies have increasingly shown that these roots also possess significant potential for various applications.” Please specify the nature of these applications or provide relevant references.

Response: Thank you for your comments. We have revised and expanded the relevant content and reorganized the structure of the paragraphs. And the following section "Zhao et al. [21] isolated and identified pseudostellarin K (PK)…" is an explanation of this sentence.

4) Table 1

It would be valuable to include the method(s) of compound identification used for each of the cyclic peptides (CPs) listed in the table.

Response: Thank you for your comments. The main structural identification methods of the CPs mentioned in the table are usually the combination of nuclear magnetic resonance and mass spectrometry, so no additional column was specially added to describe them.

5) Section 3.1 (Lines 106–116)

The authors are encouraged to provide additional details regarding the peptide isolation procedures. Specifically, did the authors employ column chromatography (CC), high-performance liquid chromatography (HPLC), or another technique? Note that the term elution is commonly associated with chromatography and is not typically used in extraction methodology; please revise the related sentences accordingly.

In my professional experience, high-speed counter-current chromatography (HSCCC) is a productive method for the preliminary separation of natural products, but it often requires a follow-up purification step using HPLC. The authors should consider including the following statement in the discussion section only if it is directly relevant to the isolation of CPs:

“High-speed countercurrent chromatography (HSCCC) has been widely applied in the separation and purification of natural products in recent years due to its advantages, including the absence of irreversible adsorption, high recovery rate, and ease of operation [26]. Han et al. [27] employed…”

Response: Thank you very much for your kind comments. We have reorganized Section 3.1 into two parts: extraction and purification. The purification methods were described in the second paragraph. The use of HSCCC in the purification of CPs was introduced.

6) Section 4.

The authors should clearly state which model was employed and provide additional methodological details about the study described by Zhao et al. [41]

Response: Thank you very much for your kind comments. We have added more details about the models and experimental methods associated with this study.

7) Section 6.

The conclusion (lines 417–424) is intriguing and presents an interesting perspective. It would be beneficial if the authors could discuss the relationship between structure–activity data and TCM-based knowledge of this herbal drug. Additionally, this part of the conclusion would be strengthened by the inclusion of references supporting this viewpoint, possibly involving similar peptides.

Response: Thank you very much for your comments. We have added a paragraph in the discussion section to describe the activity research of other peptides from Chinese medicine, suggesting that TCM could provide valuable insights for the development of bioactive peptides.

8) Comments on the Quality of English Language

There is an apparent inconsistency in the quality of English throughout the manuscript. It appears that the conclusions may have been further edited using language-enhancement tools, which likely contributes to their clarity. The authors are advised to ensure consistency across the entire text, paying particular attention to technical terminology, especially in the sections dealing with chemistry.

Response: Thank you very much for your comments. We have made our best effort to improve the English throughout the entire manuscript, particularly in the sections related to chemistry.

Round 2

Reviewer 2 Report

Comments and Suggestions for Authors

Thank you for the thoughtful revision. The manuscript is much clearer now and covers the topic well. Just a couple of suggestions before it’s ready:

1) The SAR section still feels underdeveloped. It would help to briefly emphasize the need for future SAR studies, even if current data are limited.

2) A short comparative summary (table or paragraph) of the extraction and synthesis methods could improve Section 3.

Looking forward to seeing this published after minor adjustments.

Comments on the Quality of English Language

The language has improved, but some awkward phrasing and grammatical inconsistencies remain. A final round of language editing is recommended to ensure clarity and fluency.

Author Response

1) The SAR section still feels underdeveloped. It would help to briefly emphasize the need for future SAR studies, even if current data are limited.

Response: Thank you very much for your suggestion. We have added a paragraph in conclusion to emphasize the importance of future SAR studies.

2) A short comparative summary (table or paragraph) of the extraction and synthesis methods could improve Section 3.

Response: Thank you very much for your kind comment. We have reorganized Section 3.2, and added a paragraph to summarize the preparation methods at the beginning of Section 3.